# Early intervention model for treating mood and anxiety disorders: A realist mixed-methods hypothesis test of emerging adult recovery through the mechanism of agency

**Elizabeth Osuch**[1,2,3]*, **Evelyn Vingilis**[3,4], **Michael Wammes**[1,3], **Jazzmin Demy**[5], **Carolyn Summerhurst**[6], **Justin Arcaro**[7]

1 Lawson Health Research Institute, London, Ontario, Canada, 2 FEMAP; London Health Sciences Centre, London, Ontario, Canada, 3 Department of Psychiatry, Schulich School of Medicine and Dentistry, University of Western Ontario, London, Ontario, Canada, 4 Department of Family Medicine, The Western Centre for Public Health and Family Medicine, Schulich School of Medicine and Dentistry, London, Ontario, Canada, 5 Department of Psychology, Faculty of Health, York University, Toronto, Ontario, Canada, 6 London Health Sciences Center, London, Ontario, Canada, 7 College of Medicine, American University of Antigua, University Park, Coolidge, Antigua & Barbuda

* elizabeth.osuch@lhsc.on.ca

**Data Availability Statement:** The dataset generated and analyzed for the current study are available from the Open Science Framework (OSF;

## Abstract

Early intervention treatment programs for mood and anxiety disorders are desperately needed since incidence of these is increasing. Evaluating such programs can identify which model components are helpful in providing improved outcomes. Realist evaluations discuss context-mechanism-outcome configurations to identify which interventions help whom, how, and under what circumstances. This study presents a realist configuration to evaluate an early intervention mood and anxiety program. The intervention involves personalized treatment in a shared decision-making model. The context of the model and the intervention, which uses a personalized, holistic, patient-centered approach that supports and facilitates agency enhancement within patients is described. The hypothesized mechanism of recovery is improved individual agency of the patient. Mixed methods were used to assess the proposed configuration. Illness severity measures were compared before engagement and 1–2 years after treatment onset. Results show improved functioning as well as improved symptoms, better quality of life and satisfaction with care. Individuals experienced significant functional improvement, with a large effect size. Symptoms and quality of life also improved significantly with large effect sizes. Reported satisfaction was high. Improvement in functioning was correlated with improvement in coping style but not age, number of visits, duration between timepoints or total number of traumatic exposures. Qualitative data also addressed the hypothesized mechanism of recovery. Youth identified their own engagement in care as an essential source of recovery and attributed improved agency as integral to overcoming life disruptions caused by mental illness. This realist evaluation is preliminary or pilot, and future work is needed to assess the hypothesized configuration more comprehensively and in different populations.

DOI 10.17605/OSF.IO/T57MN). Qualitative interview guides used in this study are also available at OSF as well. Details of the methods and model are expanded in the Supporting information section of this article.

**Funding:** This project was supported by the Innovation Fund of the Alternate Funding Plan of the Academic Health Sciences Centres of Ontario (INN13-008); and an un-numbered grant from the Canadian Depression and Intervention Network (CDRIN). In-kind support came from the University of Western Ontario, Schulich School of Medicine and Dentistry, the London Health Sciences Centre, the London Health Sciences Foundation, St. Joseph's Health Care, and the Lawson Health Research Institute. The funders played no role in study design, writing of this manuscript or the decision to submit for publication.

**Competing interests:** No conflict of interest, actual or apparent, exists for this study for any of the authors. Dr. Osuch. received an unrestricted research grant from Pfizer Pharmaceutical Company in the past, which was unrelated to this project. This does not alter our adherence to PLOS ONE policies on sharing data and materials.

# Introduction

Mental illnesses overwhelmingly begin in emerging adults (EAs) [1], and their burden is great due to their long duration [2, 3]. Absence of adequate treatment leads to high levels of disability adjusted life-years [4, 5]. Improving functioning in EAs is especially important to reduce the burden associated with these illnesses [6, 7] and to return EAs to health. Early intervention for mood and anxiety disorders is recognized as imperative to reduce this problem [4, 8, 9].

Given increasing need for services and insufficient system capacity [10–13], particularly after the COVID-19 pandemic [14], it is essential that effective models of early intervention are identified. Mental health systems evaluation research is a priority in the context of this challenge [11]. Depending on the stage of development and goals of the model, evaluations of early intervention programs have taken the form of needs assessments, process evaluations, outcome/impact assessments and cost-effectiveness evaluations [15]. Yet program evaluations have shown mixed results in early intervention in psychotic illness [16, 17] and in the context of general mental unwellness [6].

Recently, early intervention program evaluations have incorporated questions consistent with a realist perspective, which seeks to understand the complexity and interactions among participants, providers, environment, and the interventions themselves [18]. A realist perspective includes questions like: "what works for whom? How? And under what circumstances?" [19]. These are especially important for the evaluation of a mental health program because of the complex bio-psycho-social context of these illnesses and their treatments. Considering the realist perspective for the expansion or spread of a model is especially important so that good results are duplicated, and mixed results may be understood. As described within the realist evaluation paradigm, understanding and defining the *contexts* (Cs), *mechanisms* (Ms) and *outcomes* (Os) within any intervention are essential for creating and sustaining effectiveness at different sites and times [19]. Addressing *contexts* and *mechanisms* can assist in understanding outwardly conflictual *outcomes* within a program or across them [20], and suggests how *outcomes* may be improved by optimizing relevant *mechanisms*.

The First Episode Mood and Anxiety Program (FEMAP) provides treatment to EAs, age 16–25 (at time of screening), experiencing mood and/or anxiety concerns. We have previously conducted process- [21–23] and cost-evaluations [24, 25] of the model. We have also conducted qualitative research on EAs perspectives of treatment [26–28]. Here, in order to identify as much as possible from the model, we endeavor to describe a hypothesized *contexts-mechanisms-outcomes configuration* (CMOC) of FEMAP by utilizing both long-term quantitative and qualitative data. The goal is to better delineate how, when, and for whom the model is effective. This is relevant for replicating the model in other contexts.

# Methods

A graphic depiction of our proposed CMOC is illustrated in Fig 1. Note that this depicts a model that has evolved and grown from clinical need. Initial design was based on existing early intervention psychosis programs, with which it shares some elements [29, 30], but from which it departed according to the differing needs of the patient population.

The overarching target patient *context* of the model is EAs with moderate to severe symptomatology with primary mood/anxiety challenges, sometimes called stages 2 and 3a at the start of FEMAP engagement [31]. Youth presenting during the time of the study had an array of psychiatric challenges alone or in combination, including, but not limited to: anxiety, mood symptoms, substance use, posttraumatic stress disorder (PTSD), obsessive-compulsive disorder or personality disorder, behavioral addictions, affect dysregulation with repetitive non-suicidal self-injury or chronic suicidality, attachment challenges, eating disorders, attention

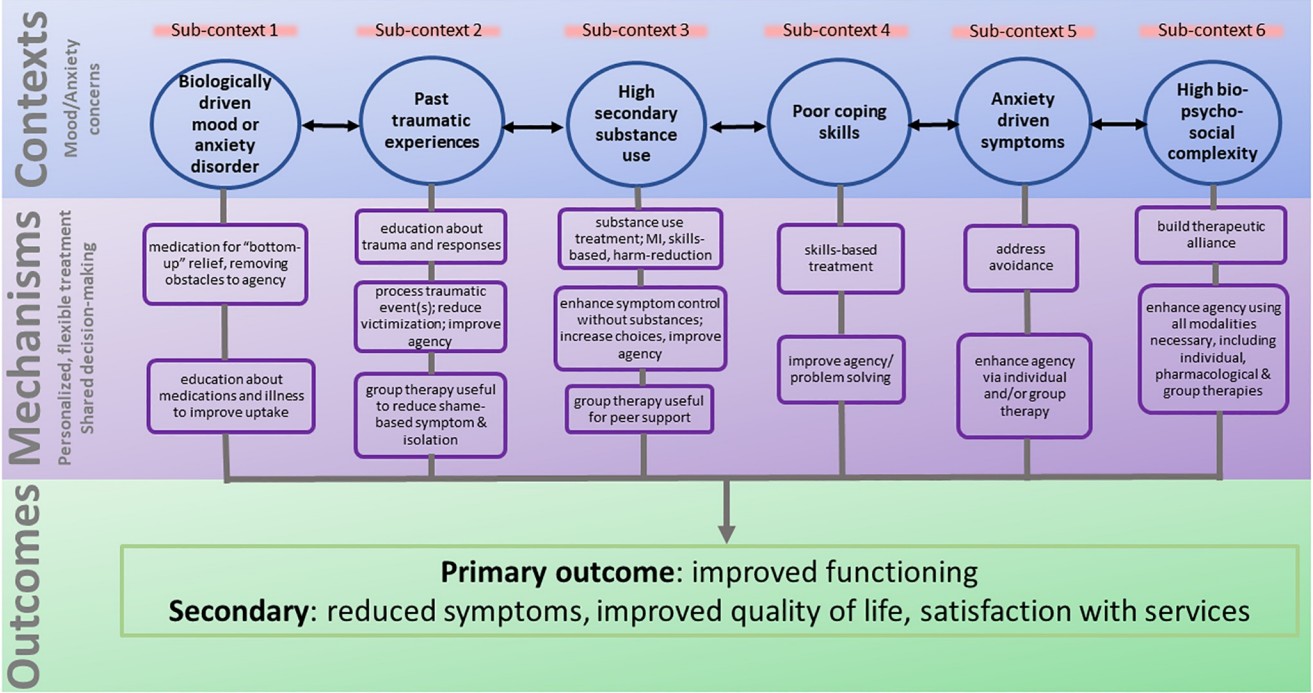

**Fig 1. Context-Mechanism-Outcome illustration of FEMAP model.** Sub-contexts indicate presumed drivers of the overall context, which is mood and/or anxiety concerns. Sub-contexts may overlap and, thus, arrows that move across CMO columns are included. Mechanisms include the intervention of personalized, flexible treatment targeted to the biopsychosocial needs of the individual, with a shared decision-making approach.

deficit hyperactivity disorder, and/or mild autism spectrum disorder. Such individuals were not excluded because these co-occurrences are common [7, 32].

The *context* for the program model excluded primary psychosis and developmental disabilities. EAs with ongoing legal charges, externalizing disorders, or extensive criminal behaviors also require different approaches and were not within the *context* of the model. EAs with over 18 months of total lifetime medication use were excluded to both avoid "doctor shopping" by individuals with existing mental health services and because EAs with psychiatric treatment starting in childhood are more appropriate for more intensive specialized services. Additional exclusionary *contexts* were major medical illnesses that affect cognition or cause mood/anxiety directly (e.g., multiple sclerosis, brain tumors, fibromyalgia, other chronic pain syndromes). Head trauma with loss of consciousness for longer than a few moments, and a primary substance use disorder (i.e., mental health concerns beginning only after a substance use disorder, by history) were also not within the *context* of the model.

The *intervention* provided within the FEMAP model, as defined and pertaining to the participants described here, involves a personalized, holistic approach that relied on a multi-disciplinary team of licensed, clinically-trained professionals skilled in an array of evidence-based intervention strategies, including psychiatrists and psychotherapists (psychology, social work, clinical counseling), practicing within the guidelines of their professional training. Psychiatrists working in the model were comfortable providing psychotherapy for integrated treatment and provide the entirety of treatment, when able. This is compatible with prior observations that splitting the treatment reduces efficiency compared with one practitioner conducting both psychopharmacology and psychotherapy [33].

The treatment model is youth-friendly, non-stigmatizing and located outside of a formal clinical hospital building. The goal is to help return EAs to their developmental trajectory to become healthy, independently functioning adults. Access is "low-barrier" by not requiring a physician referral. An in-depth bio-psycho-social intake assessment is conducted to correctly identify the most appropriate level of care for system efficiency (viz., stratified care) [34]. Assignment to a specific clinician is decided based upon expertise of clinicians for the treatment indicated, as decided in a case conference after intake assessment. For example, different clinicians have expertise in trauma treatment, substance use treatment, anxiety, bipolar disorder, obsessive compulsive disorder, eating disorders, etcetera. Patients sometimes indicate a desire to work with clinicians of a particular gender or ethnicity or express an interest in medication or no medication. To the extent possible, optimal pairings are made in order to honor the patient's preferences and increase the probability of a strong therapeutic alliance.

Individual treatment provides the foundation upon which other interventions may be added. Evidence-based treatment is offered as clinically indicated and as accepted, to potentially include individual psychotherapy, family therapy, group therapy, addictions counseling, and pharmacotherapy. Adding a treatment provider/modality is decided by the clinician and the EA, based on clinical indications and patient interest. Evaluations of *outcome* metrics are administered across multiple time-points for the sake of this research (see S1 File).

Treatment decisions are based on clinical indicators rather than strict formulaic psychotherapy steps or predetermined end-points. Within individual therapy, clinicians have the experience and expertise to engage the EA in a way that is believed to be the most clinically appropriate, and to adjust the evidence-based modality depending on the EA's engagement with, and response to, treatment. This allows for flexibility to meet the needs of the individual.

This approach, based on principals rather than an algorithm, is responsive to concerns about formulaic approaches to patients who are not in the demographic where the formulae were developed, including youth [35] and marginalized individuals [36]. In the description of DBT for incarcerated youth, Fasulo et al., remind us that the clinician must, "put aside their agenda" and focus on the needs of the patient to be useful [36]. This approach recognizes that every patient is unique, at a specific developmental level, and has distinct needs, strengths and challenges. It is incumbent upon the clinician to "enter the patient's world" therapeutically, rather than require that the patient enter the clinician's world. This approach is described in more detail in S1 File.

Group therapies adhere to guidelines as designed by co-facilitators, with adaptations based on group processes and participant responses. Groups available during the time of quantitative data collection included cognitive behavioural therapy for depression/anxiety, Stabilization Group for young women with traumatic exposure, Worry Group for generalized anxiety disorder, and Social Anxiety Group. More recently, and reflected in the qualitative data, additional groups included Dialectical Behavioural Therapy skills group, and Seeking Safety for PTSD plus substance use.

## Hypothesized mechanism

Many EAs with mood and/or anxiety disorders have missed psychosocial developmental milestones due to symptoms and/or circumstances. The "final common pathway" of such challenges may be described using the concept of agency and its impairment [37]. We use the term "agency" to mean the ability to act towards a goal over time, in spite of obstacles, to achieve that goal. This is associated with initiative and industry, per Erikson's stages of psychological development [38], and with ego strength [39]. It involves identifying realistic goals, persisting in the face of the kinds of setbacks encountered by older adolescents and young adults, and

achieving a desired result. We use "agency" rather than a "sense of agency" which is an inner belief in one's ability to act as an agent, because that may or may not accurately reflect the ability itself.

Agency is especially relevant for EAs because of the adolescent developmental stage of separation-individuation [40]. Late adolescence/early adulthood is an important time when agency development is required to transform from childhood dependency on caregivers into a self-sufficient, decision-making adult capable of enacting complex, self-initiated activities to achieve a desired goal.

Implicit in being an effective agent is the completion of multiple previous developmental milestones, which can be represented by concepts like competence, which is expected to be largely established by adolescence [39]. Consistent with the achievement of the pre-adolescent and adolescent stages of development [41], adult levels of agency depend upon the attainment of an internal locus of control [42], identity resolution [43] and, as mentioned, ego strength [39]. We focus on agency as a *mechanism*, rather than the (arguably more abstract) concepts mentioned above, because the concept of setting a goal for oneself and working steadily towards it over time in spite of challenges until it is attained is relatively easier to operationalize, both for clinicians and for patients.

In the *context* of mood and anxiety disorders, agency may be obstructed by any number of processes (sub-*contexts*; Fig 1). Some obstacles relate a lower perceived value of the goal (reward) compared with the perceived effort to achieve the goal [44]; or by difficulties with problem-solving or manifesting persistence towards the goal. These include biologically induced alterations in motivation and reward-processing and/or problems of executive functioning, and may be purely biological processes, or bio-psycho-social ones that result from past or recent traumatic events, entrenched problematic early learning, past and/or current social environmental challenges, addictions to substances or other behaviors, etcetera.

The need for personalized treatment within the FEMAP model stems from the complex developmental pathway that is involved in achieving adult levels of agency, and the unique sub-*contexts* that may have impeded this development for each individual (Fig 1).

The FEMAP model works to remove barriers to agency for willing EAs by administering needed pharmacological interventions in those with "bottom up" mental illness symptoms interested in trying medication(s). It provides psychotherapies that enhance agency by the establishment of a stable therapeutic relationship and therapeutic approaches that provide "top down" improvements through enhancing affect regulation, reducing anxious avoidance, escaping traumatization/victimization and its legacy, and providing approaches to move beyond addictive dependencies. When effectively utilized by EAs these lead to improved ego strength and coping and, consequently, improved agency.

Efforts to provide psychiatric and psychological interventions in what is commonly called the "medical model" often rely on authority- or expert-based interventions, wherein the patient is told what to do with little attempt to engage with the broader circumstances or meaning of the patient's life, or to enhance agency within that backdrop. Although the "content" of such an intervention, such as the delivery of a potentially helpful medication or psychotherapy, is often sound, the process is flawed as a *mechanism* of developing agency. Such a process leaves the patient in a passive role, rather than engaging them as an active partner in recovery. This is equally true of rigidly "delivering" a formula-driven psychotherapy intervention "onto" a patient without actively engaging them in the process and being open to adjustments based on the patient's responses. A collaborative therapeutic process is essential to the development of agency and essential to the model described. Benefits of such an approach are well known and have been described under the rubric of shared decision-making [45].

It is our hypothesis that one primary *mechanism* by which FEMAP produces positive and lasting clinical *outcomes* is by helping individuals improve their agency. As indicated, this may occur through medications, the therapeutic relationship, psychotherapy, making connections with others to reduce isolation and helplessness, by providing role models, improving communication skills, enhancing coping strategies, reducing avoidance behaviors, and/or others, as per our CMOC (Fig 1). Conversely, the model is hypothesized to not be helpful when individuals are unable to improve their agency through the therapeutic modalities mentioned, which has helped define the *contexts* in which the model is expected to be effective.

Expected short, medium and long-term *outcomes* of the FEMAP model have been provided previously in a logic model [23], an updated version of which is included in S1 File. From the realist perspective, the *primary desired outcome* we are focusing on is improved functioning. Functional impairment is the cause of disability adjusted life-years and improvement of this is, therefore, of primary importance [4]. Additionally, our prior qualitative research indicated that improved functioning was more salient to EAs than complete symptom resolution [27]. Secondary *outcomes* include reduced symptomatology, improved quality of life, and satisfaction with care.

The Human Research Ethics Board for the University of Western Ontario/Lawson Health Research Institute approved all procedures in the three protocols (1 mixed methods, 2 qualitative) involved in data recounted within this report. All willing participants signed informed consent after the studies was explained in full and any and all questions were addressed, as per the Human Research Ethics Board for the University of Western Ontario, in accordance with the Declaration of Helsinki (reference number 2023-103559-75941). The individual projects included minors (ages 16–18). The Research Ethics Board had waived the need for parental consent for each of the 3 projects. All quantitative data were collected between June 2013 and January 2020 (before the COVID-19 pandemic declaration in Canada). Quantitative data were collected in the context of 2 additional protocols, one which started on 1 December 2019 and ended 1 July 2022; the other started on 1 October 2009 and ended 1 March 2013.

This report focuses on long-term (1–2 year) *outcomes*. Initial questionnaires were administered after a telephone screen during which the EA requested services, before in-person contact with FEMAP was made (pre-treatment; PreT). They were repeated at the time of the Intake (TI) and at long-term follow up between 1 and 2 years after treatment onset (TL/T) regardless of whether the patient was still actively in treatment. The PreT data-collection point was only added after the first 91 patients were recruited, when it was realized that the intake process itself was associated with reported increased hopefulness and symptom reduction. PreT was added to obtain a more accurate baseline. For the 91 participants without PreT data, TI data were substituted for PreT data. Since TI represents an improved state compared with PreT (confirmed in the cohort with both time-points), this substitution could only reduce the positive results and, therefore, could not contribute to a type I error. Not all measures were measured at PreT; some were first evaluated at TI. That time-point was used for those variables, as indicated. Additional details of questionnaires and timepoints are provided in S1 File.

Study data were collected and managed using the Research Electronic Data Capture Application (REDCap) tool [46, 47] hosted at Lawson Health Research Institute. Participants were compensated fifty dollars for their time and travel at TL/T.

Questionnaires had been previously validated. Some were modified, as described below. The primary *outcome* variable of patient functioning was evaluated with the Sheehan Disability Scale (SDS) [48], which measures functional impairment across 3 domains (work/school, social life, family life/home responsibilities). The SDS also queries "days lost" and "days under-productive" due to symptoms. Secondary *outcome* variables were measured using: The Montgomery-Åsberg Depression Rating Scale, Self-Report [49]; the Anxiety Sensitivity Index-3

[50]; and the Quality of Life Enjoyment and Satisfaction Questionnaire-Short Form [51]. We modified the scoring of this last scale to exclude items about prescribed medication and sexual satisfaction since these items often did not apply, were left blank, and would therefore have substantially reduced usable total scores (see details in S1 File).

Additionally, patients completed a Patient Satisfaction Questionnaire (PSQ), at TL/T [52]. This 20-item measure ranged from 0–100 points with higher scores indicating greater satisfaction. Originally from the United States, we adapted it to the FEMAP population (viz., a socialized health care system, mental healthcare). This revised questionnaire in provided in S1 File. Some PSQ questions were not applicable to all patients (medication; parking), so by face-validity, a 3-item subscale was developed using only 3 questions: "I liked the services I received here", "If I had other choices, I would still get services from this agency" and, "I would recommend this agency to a friend or family member". This produced a 0-15-point PSQ Subscale.

Additional questionnaires included the Trauma History Questionnaire [53], which evaluated number of different types of traumatic experiences at TI and TL/T; and the Brief Cope [54], which included 8 adaptive strategies (0–64 points) and 6 maladaptive strategies (0–48 points), also administered at TI and TL/T. Total scores for the Brief Cope were calculated as adaptive sum minus maladaptive sum, resulting in scores between -48 and +64, with higher scores indicating more adaptive coping. These questionnaires were expected to influence *outcomes* and provide important background to either enforce our hypothesis of agency as a *mechanism* of recovery or lead to alternative hypotheses.

Qualitative results were from prior in-depth, semi-structured interviews with patients at FEMAP, published [26–28] or in preparation for publication. The qualitative results from these projects were reviewed to identify themes relevant to the *mechanism* theorized in the proposed CMOC.

## Statistical analysis

Analyses employed the Statistical Package for the Social Sciences (SPSS), version 29.0.2.0. Missing item scores were prorated (i.e., individual mean substitution) when ≤10% of total items were missing completely at random per SPSS's missing value analysis. Individual questionnaires not meeting these criteria were removed from specific analyses. For the SDS, participants could indicate that they were neither working nor in school; in such cases the remaining two scores were averaged in place of the third value. Additionally, a total score of days lost plus days underproductive (SDS Days L/U) was created by summing these individual scores.

Repeated measures analyses within subjects using the general linear model were conducted for each variable independently to reduce missing data across variables. Effect sizes were estimated using eta-squared. Quantitative variables were normally distributed, with the exception of the PSQ and the SDS Days L/U. The PSQ was skewed to the high end of satisfaction; the SDS Days L/U was bimodal. Variables potentially interacting with *outcomes* included age, number of days between PreT and TL/T, total number of clinical visits between these timepoints, Brief Cope score, and Trauma History Questionnaire score. Non-parametric statistics were used for analyzing non-normally distributed variables. Correlations were 2-tailed and Bonferroni-corrected for multiple comparisons for each analysis.

## Results

The following quantitative and qualitative results are provided to assess the proposed CMOC.

## Quantitative

449 participants completed data at Pre-T and TL/T. The mean age at entry was 19.1 years (SD = 2.5, range = 16–25). Demographics at time of entry showed that 336 (75%) were female and 113 (25%) were male; 316 (70.5%) were enrolled in school, and 223 (50%) were employed either part-time or full-time. Only 57 (13%) were neither employed nor enrolled in school.

The median total number of clinical treatment appointments with FEMAP was 15 (mean = 16.41, SD = 11.45, range = 1–73), including all group and individual sessions. There were 2 modes, 5 session and 15 sessions. The median total number of days between PreT and TL/T was 592.3 (mean = 593.0, SD = 90.20, range = 419–1022 days), which included the time between PreT and treatment onset, which accounted for durations longer than 730 days. The largest of these numbers represent delays on the part of the EAs to engage in treatment rather than service system delays.

Over the course of the study, 348 (77.5%) of participants had a psychiatric consultation and most were treated by that psychiatrist; 105 (23.4%) saw a psychologist for individual psychotherapy; 82 (18.3%) were involved in group therapy; 63 (14.0%) saw an addictions counselor; 61 (13.6%) saw a social worker psychotherapist/case manager, and 41 (9.1%) saw a family therapist. Total number of different types of treatment providers seen by each patient are shown in Fig 2. Over 60% of participants only engaged with one type of provider over the course of treatment, as decided clinically by the client/clinician pair.

Repeated measures analysis showed significant reduction in total functional impairment over the course of study, as well as improvements in depression, anxiety sensitivity and life-satisfaction. Each of these demonstrated large effect sizes (Table 1). Functional improvement and depression symptom reduction were also clinically significant [55, 56].

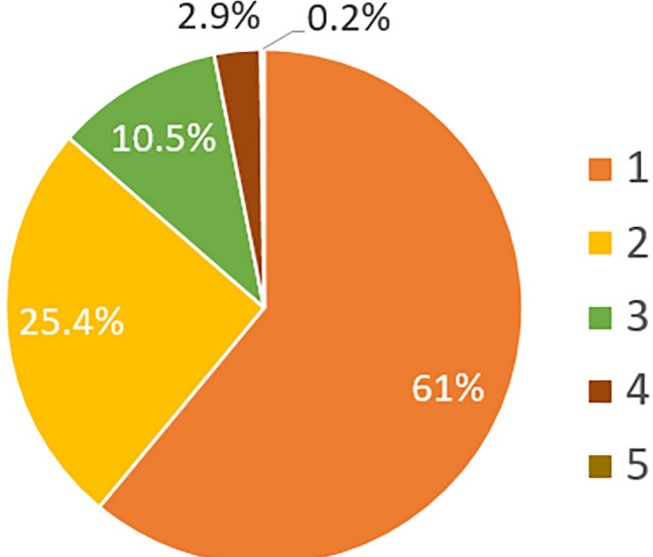

**Fig 2. Number of different treatment providers/treatments engaged by individual clients, to include six possibilities: Psychiatrist, psychologist, group therapy, addictions counselor, social worker psychotherapist/case manager, and family therapist.**

**Table 1. Behavioral scores comparing primary and secondary outcomes of the Contexts-Mechanisms-Outcomes Configuration before treatment (PreT) and 1–2 years after treatment onset (TL/T).**

| | Before Treatment (PreT) | | Follow-up (TL/T) | | Repeated measures comparison | | | |
|---|---|---|---|---|---|---|---|---|
| | Mean | SD | Mean | SD | *df* | *F*-statistic | *p*-value | Eta-squared |
| **Primary outcome variable†** | | | | | | | | |
| **SDS (n = 448)** | 18.99 | 6.40 | 10.72 | 7.55 | 447 | 421.076 | $< .001^*$ | .489 |
| **Secondary outcome variables†** | | | | | | | | |
| **MADRS (n = 448)** | 13.37 | 4.08 | 8.88 | 4.85 | 447 | 332.617 | $< .001^*$ | .427 |
| **ASI (n = 447)** | 34.68 | 14.38 | 25.74 | 16.02 | 447 | 157.877 | $< .001^*$ | .261 |
| **Q-LES-Q (n = 449)** | 34.42 | 7.86 | 39.57 | 8.61 | 448 | 127.952 | $< .001^*$ | .222 |

SD = standard deviation, SDS = Sheehan Disability Scale, MADRS = Montgomery-Åsberg Depression Rating Scale, ASI = Anxiety Sensitivity Index, Q-LES-Q = Quality of Life Enjoyment and Satisfaction Questionnaire

*Statistically significant at alpha $< .0125$ after Bonferroni correction.

†Lower scores on the SDS, MADRS, and ASI indicate less functional impairment, depressive symptoms, and anxiety sensitivity, respectively. Lower scores on the Q-LES-Q indicate less satisfaction with life. SDS$> = 12$ indicates functional impairment; SDS$<12$ indicates no functional impairment

*Post-hoc* analysis of subgroups indicated no significant difference in SDS outcome by sex, THQ grouping (1 or fewer endorsed type of traumatic exposure versus more than 1) or whether or not the patient was referred by a physician.

By Wilcoxon Signed Rank Test, SDS Days L/U was also significantly reduced at TL/T (median = 2.0 days) from PreT (median = 5.0 days) (n = 443, Test Statistic = 11990.5, Standardized Test Statistic = -11.061, p < .0005).

The median total score on the PSQ was 80 (mean = 78.8, S.D. = 13.44, range 20–100). For the PSQ 15-point subscale the median was 13 (mean = 12.7, S.D. = 2.50, range 0–15). 93% patients rated FEMAP services more positively than neutral on the total scale and 37.6% (169 patients) gave FEMAP the highest possible score on the PSQ Subscale.

Table 2 shows correlations among primary and secondary outcome variable changes from PreT to TL/T.

Correlation coefficients among age, number of days between timepoints, total number of clinical visits, coping style, and trauma exposure are shown in Table 3. Better functional *outcomes* were correlated with more adaptive coping style at TL/T. Age, number of days between data collection timepoints, number of clinical visits and THQ scores were not associated with functional *outcome* after correction for multiple comparisons. Total number of clinical visits was correlated with total THQ score.

## Qualitative

In a prior qualitative evaluation of treatment at FEMAP, patients indicated 4 themes associated with improvement, including feeling empowered to actively affect change in their life, or agency [27]. They reported learning skills so that ". . .you know what's happening, and you know how to pull up instead of taking a nose-dive" (male, age 25) [27].

In another qualitative investigation of expectations versus experiences of treatment, a majority of patients indicated that strengthening agency was part of their treatment experience [28]. They drew a contrast between this and their expectations of a more directive or authoritarian treatment experience, as well as with the expectation of a "quick fix" that involved less of their participation [28]. As one individual explained, "You can kind of be shown the door, but you have to open it yourself" (male, age 22) [28].

**Table 2. Correlations among changes in primary outcome variables (functional impairment) and depression, anxiety and quality of life satisfaction from pre-treatment to long-term follow-up.** Correlation between change in functional impairment and patient satisfaction is also shown.

|  |  | SDS | MADRS | ASI | Q-LES-Q | PSQ Total |
|---|---|---|---|---|---|---|
| **SDS** | Pearson Correlation | — |  |  |  |  |
|  | N | 448 |  |  |  |  |
| **MADRS** | Pearson Correlation | .640* | — |  |  |  |
|  | Sig. (2-tailed) | < .001 |  |  |  |  |
|  | N | 447 | 448 |  |  |  |
| **ASI** | Pearson Correlation | .297* | .427* | — |  |  |
|  | Sig. (2-tailed) | < .001 | < .001 |  |  |  |
|  | N | 446 | 446 | 447 |  |  |
| **Q-LES-Q** | Pearson Correlation | -.567* | -.701* | -.302* | — |  |
|  | Sig. (2-tailed) | < .001 | < .001 | < .001 |  |  |
|  | N | 448 | 448 | 447 | 449 |  |
| **PSQ Total** | Spearman Correlation | -.129* | -.180* | -.062 | .268* | — |
|  | Sig. (2-tailed) | .006 | < .001 | .089 | < .001 |  |
|  | N | 448 | 448 | 447 | 449 | 449 |

* Correlation is significant at <0.01 (2-tailed), Bonferroni corrected for 5 variables.

SDS = Sheehan Disability Scale, MADRS = Montgomery-Åsberg Depression Rating Scale, ASI = Anxiety Sensitivity Index, Q-LES-Q = Quality of Life Enjoyment and Satisfaction Questionnaire, PSQ Total = Patient Satisfaction Questionnaire, total score

**Table 3. Correlations among potential contributing variables.**

|  |  | SDS Change | Age at TL/T | Days, PreT to TL/T | # Clinical Visits | BC at TL/T | THQ at TL/T |
|---|---|---|---|---|---|---|---|
| **SDS Change (PreT to TL/T)** | Spearman's rho | — |  |  |  |  |  |
|  | N | 448 |  |  |  |  |  |
| **Age at TL/T** | Spearman's rho | -.036 | — |  |  |  |  |
|  | Sig. (2-tailed) | .445 |  |  |  |  |  |
|  | N | 448 | 449 |  |  |  |  |
| **Days, PreT to TL/T** | Spearman's rho | -.043 | -.094 | — |  |  |  |
|  | Sig. (2-tailed) | .360 | .046 |  |  |  |  |
|  | N | 448 | 449 | 449 |  |  |  |
| **# Clinical Visits** | Spearman's rho | .019 | .107 | -.022 | — |  |  |
|  | Sig. (2-tailed) | .692 | .023 | .643 |  |  |  |
|  | N | 448 | 449 | 449 | 449 |  |  |
| **BC at TL/T** | Spearman's rho | -.372* | .090 | .024 | .073 | — |  |
|  | Sig. (2-tailed) | < .001 | .057 | .606 | .124 |  |  |
|  | N | 446 | 447 | 447 | 447 | 447 |  |
| **THQ at TL/T** | Spearman's rho | .090 | .074 | .113 | .144* | .019 | — |
|  | Sig. (2-tailed) | .064 | .128 | .020 | .003 | .700 |  |
|  | N | 422 | 423 | 423 | 423 | 422 | 423 |

* Correlation is significant at <0.008 level (2-tailed), Bonferroni corrected for 6 variables.

SD = standard deviation; SDS = Sheehan Disability Scale; PreT = timepoint before entering FEMAP; TL/T = Long-term follow-up timepoint; BC = Brief Cope, THQ = Trauma History Questionnaire

In the course of the qualitative study on human dignity in the context of youth mental health care and FEMAP, the following quotes about the FEMAP model are salient (manuscript in preparation):

*And, um, for me, they've, um, gotten the medication to help my bipolar disorder be manageable, um, to the point where I can go to work every day, I can keep my house relatively clean. I can keep myself based [sic]. I'm not putting myself in harm's way anymore. They have provided me with different counselors for different things-like I said addictions, trauma, family counseling-um, to the point where I have been able to kind of understand my own worth. . . . Understand. . ., where I have power and where I can, you know, make my own decisions and things like that.*

(Participant 27–007, female, age 24)

*I just finished a group, like a therapy group. . ., centered around healing PTSD in relation to substance use issues. . ., and the practitioners running it were really meeting us with compassion, and it really felt that our perspectives, uh, were valuable to them and it didn't feel that we were being 'handled'. It felt more that we were being, that we were working together towards recovery, rather than, um, them just knowing what's best for us and imposing it on us.*

(Participant 27–010, female, age 24).

## Discussion

This study was an initial exploration of the FEMAP model from a realist perspective, focusing on a specific hypothesized CMOC. Our interest was to explore the treatment *contexts* in relation to *outcomes*, and whether agency, our hypothesized *mechanism*, was in accordance with those *outcomes*.

The long-term quantitative *outcomes* analyzed here identified that patients improved in both primary and secondary measures, in a highly intercorrelated pattern. Specifically, over the course of study, FEMAP patients improved in functioning, had reduced symptom severity and improved life-satisfaction; they were highly satisfied with the care they received, on average.

The absence of correlation between functional *outcomes* and age, number of visits, or time between data-points indicates that these not strongly related. Additionally, the negative *post hoc* findings of subgroup differences based on sex, trauma exposure intensity and physician referral (suggesting more involvement of that practitioner) indicated that these variables did not have a major impact on *outcomes*. This is consistent with our hypothesis about the *mechanism* involved in improvement, which was not directly related to biological age, sex, trauma exposure, primary care use, clinical intensity or merely to the passage of time, but rather related to a growth in personal agency.

There was a correlation between coping style and *outcomes*, but not between number of different types of traumatic exposures and *outcomes*. The former finding is consistent with our proposed *mechanism* of recovery, since agency and adaptive coping are expected to be associated. The lack of relationship between functional *outcome* and THQ score or category (in the *post hoc* analysis) was unexpected, since the traumatized population tends to be more complex, although a metric of total traumatic exposure, rather than number of *different types* of traumatic exposures, may have yielded different results. THQ score was correlated with number of

clinical visits, however, implying greater intensity of treatment related to this variable. This finding supports the model as responsive to individual variation in patient needs.

Median number of clinical contacts was 15, which is more than typically allowed under many medical insurance policies. There were two modes for this variable, however, 5 and 15 sessions, confirming that there were different care trajectories for patients within the cohort. Total number of sessions included group therapy, which often ran for 8–12 sessions. In the current cohort, less than 20% of patients participated in group therapy because that was the last modality to be developed within the model. It is anticipated that there will be more involvement in groups going forward, which may change the number of clinical contacts per patient, presumably leading to an increase in this number. This will be important to monitor, together with *outcomes*, to optimize model efficiency. Notably, experience at FEMAP showed that EAs were reluctant to start their treatment process with groups–they almost uniformly would not agree to group therapy without first engaging in individual treatment (clinical observation), though that may not hold in other population samples.

Although the treatment team was multidisciplinary, a majority of patients only saw one practitioner over the course of their treatment based on their clinical need and influenced by patient preference. Differences in clinical outcomes of split versus integrated treatment in mental health care have not been adequately evaluated [57]. Yet cost comparisons have found integrated care to be equivalent or less costly than split care [33, 58]. It is our contention that matching the individual EA with a clinician best suited for their needs reduces cost of care. Additionally, training and supporting clinicians to know when to "switch gears" on the approach to suit the needs of the patient (psychopharmacology, various psychotherapies) is expected to produce the most efficacious strategy for improvement in the *contexts* associated with this model.

Consistent with our hypothesized CMOC, patients' qualitative data indicated that they perceived the strengthening of their individual agency or empowerment to be a key component of recovery. They identified that the process of engaging as an active member of their treatment team and being involved as an agent in their own treatment was integral to their recovery. This is a critical aspect in the development and spread of the model. Existing research on shared decision-making in mental health care also supports this approach [45, 59, 60].

It is important to note that although participants indicated that they expected a more top-down authoritative process of treatment, where they would be told what to do to get better, our data showed that the reality of their experience at FEMAP, which was the opposite, was associated with recovery and was also preferred [28]. In the nuanced conversations from the qualitative data, participants indicated that they gained much by being actively engaged in a way that helped them develop their own "voice", volition or agency to act effectively on their own behalf.

Realist evaluations of similar models that provide mental health care treatment are scarce, though some protocols for doing this have been published [61–63]. Hopefully, those forthcoming studies and the results reported here will encourage others to conduct similar evaluations of mental health services for youth and EAs.

Limitations: Limitations of this study's reported quantitative *outcomes* include threats to internal validity of history, maturation and regression to the mean associated with self-report questionnaires. Not all patients completed long-term follow up data and self-selection may have occurred. There was no control group who were not in the program, so improvement cannot unequivocally be attributed to FEMAP. FEMAP is located in a specific region and time, with unique demographics, cultural context and other variable, limiting generalizability of these results. The use of the COMC paradigm in realist evaluation is specifically intended to recognize such influences so that additional iterations of a model may account for such

differences. Future studies should investigate the "fit" of the COMC structure described here in other early intervention mood/anxiety programs. Additionally, future studies should investigate the longer-term "durability" of the improvement identified here with extended measurement timelines. Future studies could also investigate cognitive factors and biomarkers to identify if predictions of outcomes can be detected.

## Conclusions

The data reported here in the framework of a CMO realist configuration suggest that efforts to duplicate the "contents" of treatment of this model will not be as effective as replicating its "processes" or *mechanism*, which includes engagement with the EA as an active and primary member of the treatment team interacting with clinicians who have the expertise to adjust the evidence-based treatments to the developmental and clinical needs of the patient. These skills are taught to a variety of clinical professionals, from psychiatrists to social workers, psychologists, registered psychotherapists, and clinical counselors. The model is in contrast to the implementation of formulaic treatment approaches that suggest identical treatment for patients with identical diagnoses with little input from the patient. Such models often include a maximum number of allowable sessions or time-interval between entry and discharge. Our COMC acknowledges that there are many variables, or sub-*contexts*, that influence individuals in both the development of their illnesses and their recovery via our proposed *mechanism* of improved agency. Therefore, a diverse skill set in both individual clinicians and the team is required, as found in multidisciplinary team approaches.

This study showed preliminary support for our hypothesized realist CMOC. The model of FEMAP, with its personalized, holistic patient-centered approach for EAs with mood and anxiety disorders and co-occurring conditions, supported and facilitated the *mechanism* of agency enhancement within patients leading to the positive *outcomes* described. This realist evaluation is preliminary or pilot, and future work to assess the hypothesized configuration more comprehensively and in different populations is needed.

## Supporting information

**S1 File.** A. Protocol. a. Objectives and Aims. b. Methodology. i. Overview. ii. Screening. iii. Intake & Assessment. iv. Data Collection–Quantitative. v. Data Collection–Qualitative. B. Treatment Model. C. References. Patient Satisfaction Questionnaire for FEMAP.
(DOCX)

## Acknowledgments

The authors would like to acknowledge the entire FEMAP team of clinicians, researcher, administrators, and patient/participants for their contribution to this research, without which it would not be possible.

## Author Contributions

**Conceptualization:** Elizabeth Osuch, Evelyn Vingilis.

**Data curation:** Michael Wammes, Jazzmin Demy, Carolyn Summerhurst, Justin Arcaro.

**Formal analysis:** Elizabeth Osuch.

**Funding acquisition:** Elizabeth Osuch.

**Investigation:** Elizabeth Osuch.

**Methodology:** Elizabeth Osuch, Evelyn Vingilis, Michael Wammes, Carolyn Summerhurst.

**Project administration:** Elizabeth Osuch, Michael Wammes.

**Software:** Michael Wammes, Jazzmin Demy, Justin Arcaro.

**Supervision:** Elizabeth Osuch.

**Validation:** Elizabeth Osuch, Evelyn Vingilis, Jazzmin Demy, Carolyn Summerhurst, Justin Arcaro.

**Writing – original draft:** Elizabeth Osuch.

**Writing – review & editing:** Elizabeth Osuch, Evelyn Vingilis, Michael Wammes, Jazzmin Demy, Carolyn Summerhurst, Justin Arcaro.

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
