## [Decision Letter · Decision Letter 0]

2 Apr 2024

PMEN-D-23-00072

Early intervention model for treating mood and anxiety disorders: a realist mixed-methods hypothesis test of emerging adult recovery through the mechanism of agency

PLOS Mental Health

Dear Dr. Osuch,

Thank you for submitting your manuscript to PLOS Mental Health. After careful consideration, we feel that it has merit but does not fully meet PLOS Mental Health’s publication criteria as it currently stands. Therefore, we invite you to submit a revised version of the manuscript that addresses the points raised during the review process.

We look forward to receiving your revised manuscript.

Kind regards,

Sasidhar Gunturu, MD

Academic Editor

PLOS Mental Health

Journal Requirements:

1. Please send a completed 'Competing Interests' statement, including any COIs declared by your co-authors. If you have no competing interests to declare, please state "The authors have declared that no competing interests exist". Otherwise please declare all competing interests beginning with the statement "I have read the journal's policy and the authors of this manuscript have the following competing interests:"

2. Please provide a/amend your detailed Financial Disclosure statement. This is published with the article. It must therefore be completed in full sentences and contain the exact wording you wish to be published.

Additional Editor Comments (if provided):

Reviewers' comments:

Reviewer's Responses to Questions

**Comments to the Author**

1. Does this manuscript meet PLOS Mental Health’s publication criteria? Is the manuscript technically sound, and do the data support the conclusions? The manuscript must describe methodologically and ethically rigorous research with conclusions that are appropriately drawn based on the data presented.

Reviewer #1: Yes

Reviewer #2: Yes

2. Has the statistical analysis been performed appropriately and rigorously?

Reviewer #1: Yes

Reviewer #2: Yes

3. Have the authors made all data underlying the findings in their manuscript fully available (please refer to the Data Availability Statement at the start of the manuscript PDF file)?

Reviewer #1: No

Reviewer #2: Yes

4. Is the manuscript presented in an intelligible fashion and written in standard English?

Reviewer #1: Yes

Reviewer #2: Yes

5. Review Comments to the Author

Reviewer #1: Review

1. What are the main claims of the article and what is their significance for the discipline?

The main claim of the study is that the improvement of the patient's individual "agency", their ability to act independently and make decisions, is a key mechanism for recovery.

The study's quantitative and qualitative findings demonstrate significant improvements in functioning, symptom reduction, quality of life, and satisfaction with treatment among participants. These results not only support the efficacy of the proposed early intervention model but also underscore the importance of personalized, patient-centered treatment that promotes their autonomy and participation in the recovery process.

It provides empirical evidence on the effectiveness of a treatment model that emphasizes patient agency, something relevant due to the increasing prevalence of these conditions and the need for more effective and personalized treatment approaches.

2. Are the claims appropriately placed in the context of previous literature? Have the authors treated the literature fairly?

The authors have contextualized their claims within the framework of existing literature in an appropriate and fair manner. This approach is reflected in several key areas of the manuscript:

Contextualization of the need for early intervention: The authors review the literature, highlighting the growing incidence of mood and anxiety disorders, and the critical importance of early intervention. Citing previous studies, they establish a solid foundation for the urgent need to develop and evaluate effective treatment programs for this population.

Realistic evaluation and personalized treatment approach: The mixed realistic methodology used by the authors is justified through a discussion on the limitations of conventional treatment approaches, relying on previous literature that advocates for more nuanced assessments considering the individual context, mechanisms of action, and outcomes. This discussion not only situates their claims within the context of previous research but also persuasively argues in favor of their methodological approach.

Agency as a recovery mechanism: The central hypothesis that the individual patient's agency is a key mechanism for recovery is placed within a well-established theoretical framework. The authors review relevant literature on the development of agency in young adults and its importance in the treatment of mental disorders, convincingly arguing that their study addresses a gap in existing research.

Fairness in the treatment of literature: Throughout the article, the authors treat the existing literature equitably, acknowledging both advances and limitations of previous studies. Their review of the literature not only validates the need for their study but also demonstrates a deep understanding of the field, showing respect for previous work while arguing for the need for their unique approach.

Discussion and conclusions: In their discussions, the authors compare their findings with those of previous studies, acknowledging the contribution of their work to the existing body of knowledge and suggesting directions for future research. This balanced approach underscores their commitment to a fair and constructive discussion of relevant literature.

3. Do the data and analysis fully support the claims? If not, what additional evidence is needed?

The review of the provided data and analysis suggests that they largely support the central claims of the study regarding the efficacy of this treatment approach. However, there are aspects that could benefit from additional evidence or deeper analysis.

Aspects Supporting the Claims

Significant Improvements in Participants: The quantitative results showed significant improvements in functioning, symptoms, quality of life, and treatment satisfaction of the participants. This provides a solid foundation for the claim that the early intervention model is effective for this population,.

Qualitative Data Reinforcing the Agency Mechanism: The qualitative evidence obtained from interviews with participants supports the claim that strengthening the patient's agency is a crucial mechanism for recovery. Participants' testimonies illustrate how active engagement in their treatment was fundamental to their improvement.

Aspects Requiring Additional Evidence

Control of Confounding Variables: Although the study shows significant improvements, the lack of a control group makes it difficult to attribute these improvements exclusively to the treatment model. Future studies with a controlled design would be beneficial to rule out other factors that could influence the results.

Analysis of Subgroups: The study could delve into how different subgroups (for example, based on the severity of the condition, gender, or the presence of concurrent conditions) respond to the treatment. This would help better understand for whom the model is most effective and why.

Long-term Follow-up: While the study presents data 1-2 years after the start of treatment, additional research including longer follow-ups could provide valuable information about the durability of the treatment effects.

Comparison with Other Treatment Models: To further strengthen the study's claims, it would be useful to compare the outcomes of the early intervention model with those of other established treatment approaches. This would help contextualize its effectiveness in relation to available alternatives.

Additional Outcome Measures: Incorporating additional outcome measures, such as cognitive functioning or biomarkers of stress, could provide a more holistic understanding of how the treatment affects patients.

4. If a protocol is provided, for example for a randomized controlled trial, are there significant deviations from it? If so, have the authors adequately explained why the deviations occurred?

The reviewed article does not specifically mention the implementation of a protocol for a randomized controlled trial (RCT) within its methodology. Instead, it describes a mixed-methods study that evaluates the efficacy of an early intervention program for mood and anxiety disorders, focusing on the mechanism of agency for recovery. Given this focus, the question of deviations from an RCT protocol does not directly apply to the design of the presented study.

However, the study does address the implementation of its research design and methodology in detail, including the justification for its mixed-methods approach, participant selection, and the interventions carried out. These elements are critical for understanding how the research was conducted and for assessing its rigor and validity.

Key Methodological Elements Mentioned:

Participant Selection: The criteria for inclusion and exclusion of study participants are detailed, which is essential to ensure that the study population is representative of those for whom the intervention program is designed.

Interventions Performed: The personalized, patient-centered treatment approach is described, including shared decision-making and the holistic approach that promotes the patient's agency as a mechanism of recovery.

Data Collection and Analysis: The study uses a combination of quantitative and qualitative methods to assess the outcomes of the intervention, allowing for a richer and more multifaceted understanding of its effectiveness.

5. PLOS Mental Health encourages authors to publish detailed protocols and algorithms as online supporting information. Does any particular method used in the manuscript warrant such treatment?

Given the innovative nature of the study and the focus on personalized and patient-centered treatment, several aspects of the method used justify the publication of detailed protocols and algorithms as online supporting information, in line with the recommendations of PLOS Mental Health.

Methods Justifying Detailed Publication:

Mixed Realist Evaluation: The study employs a mix

---

## [Editor Report · Decision Letter 1]

4 Jun 2024

Early intervention model for treating mood and anxiety disorders: a realist mixed-methods hypothesis test of emerging adult recovery through the mechanism of agency

PMEN-D-23-00072R1

Dear Dr Osuch,

We are pleased to inform you that your manuscript 'Early intervention model for treating mood and anxiety disorders: a realist mixed-methods hypothesis test of emerging adult recovery through the mechanism of agency' has been provisionally accepted for publication in PLOS Mental Health.

Best regards,

Sasidhar Gunturu, MD

Academic Editor

PLOS Mental Health